# Ultrashort X-ray Free Electron Laser Pulse Manipulation by Optical Matrix

**Kai Hu** [1,2], **Ye Zhu** [2], **Zhongmin Xu** [2], **Qiuping Wang** [2], **Weiqing Zhang** [2,3] **and Chuan Yang** [2,4,*]

1   National Synchrotron Radiation Laboratory, University of Science and Technology of China, Hefei 230029, China
2   Institute of Advanced Science Facilities, Shenzhen 518107, China
3   State Key Laboratory of Molecular Reaction Dynamics, Dalian Institute of Chemical Physics, Chinese Academy of Sciences, Dalian 116023, China
4   College of Science, Southern University of Science and Technology, Shenzhen 518055, China
*   Correspondence: yangc@mail.iasf.ac.cn

**Abstract:** Free electron laser (FEL) is capable of producing ultra-short X-ray pulses. The estimation of X-ray pulse propagation is the key process of X-ray FEL beamline design. By using the Kostenbauder matrix approach, the evolution of an ultra-short pulse in a beamline system can be calculated. Therefore, it is of significant importance to investigate the Kostenbauder matrices of different kinds of X-ray optics. In this work, we derive a unified $6 \times 6$ optical matrix to describe various kinds of X-ray optical elements, including varied-line-spacing (VLS) toroidal grating, VLS spherical grating, VLS cylindrical grating, VLS plane grating, toroidal grating, spherical grating, cylindrical grating, plane grating, toroidal mirror, spherical mirror, cylindrical mirror, and plane mirror. These optics are usually adopted in soft X-ray regime. We apply this method to describe the transverse focusing, pulse front tilt, and pulse stretching after an X-ray pulse going through a VLS plane grating monochromator (VLS-PGM). We also use this approach to simulate a grating compressor which can be used to compress chirped soft X-ray pulse. This work is helpful in the design and optimization of X-ray beamline systems.

**Keywords:** free electron laser; pulse propagation; dispersion; X-ray optics

## 1. Introduction

X-ray free electron laser (XFEL) can generate ultrashort X-ray pulses with extremely high intensities [1,2]. The XFEL pulses are manipulated by X-ray optics in the beamline, such as focusing, collimation, compression, and monochromatization, before being transported to the experimental endstations. In soft X-ray regime, FEL beamline systems usually adopt reflective optics with grazing incidence angles, such as plane mirror, KB mirror [3–5], and varied-line-spacing (VLS) grating, which are components of monochromators and spectrometers [6–10]. To evaluate the X-ray beamline systems, several software packages have been developed, such as Shadow [11], SRW [12], HYBRID [13], xrt [14], and MOI [15]. These packages can effectively describe the spatial distribution of an X-ray beam after going through a beamline.

With the development of FEL facilities, the pulse duration of XFEL can now reach femtoseconds and attoseconds. To characterize the spatiotemporal properties of FEL pulses going through a dispersive X-ray beamline, scientists have to consider the spatiotemporal effects of XFEL pulses, such as spatial chirp, angular dispersion, pulse front tilt, pulse stretching, pulse compression, and so on. However, there is still a lack of simulation tools to estimate the spatial and temporal properties of XFEL pulse propagation in X-ray beamlines.

Pulse propagation in dispersion systems can be analyzed using the Fourier transform method [16,17], which is essentially a time-dependent mode decomposition (TDMD) method. We discussed the application of TDMD to propagate 3-dimensional XFEL pulses

in beamlines [18]. Here, we will discuss another method for pulse propagation, namely the Kostenbauder-matrix (**K**-matrix) method. This approach can be used to describe the propagation of an ultrashort pulse going through a linear optical system. It was first presented by Kostenbauder in [19]. Then, Akturk et. al. developed a general theory to characterize spatiotemporal properties of ultra-short pulses by using **K** matrices [20]. Afterwards, Marcus extended **K** matrices from 4 × 4 to 6 × 6 in [21]. By using the method of **K** matrices, the spatiotemporal properties mentioned above can be well estimated. Although the framework of pulse propagation by using **K** matrices has been established, the **K** matrices of X-ray optical elements adopted in X-ray beamline have not been investigated yet.

In this paper, the **K** matrices of different types of X-ray optics are investigated, and the applications of X-ray pulse propagation in X-ray optical systems are studied. The paper is organized as follows. We first review pulse propagation by using **K** matrices in Section 2, including the definition and formulation of **K** matrices, pulse propagation in real space, and Wigner phase space. In Section 3, we derive a unified **K** matrix to describe different types of X-ray optics. This unified matrix nominally belongs to VLS toroidal grating, and can further reduce to describe VLS spherical grating, VLS cylindrical grating, VLS plane grating, toroidal grating, spherical grating, cylindrical grating, plane grating, toroidal mirror, spherical mirror, cylindrical mirror, and plane mirror. Applications of our method for VLS plane grating monochromator (VLS-PGM) and grating pulse compressor are discussed in Section 4. We first apply our model to simulate pulse stretching, pulse front tilt and pulse focus after a VLS plane grating monochromator (VLS-PGM), and the benchmark is performed by using Shadow and SRW. Then, we apply our method to estimate X-ray chirped pulse compression by a grating compressor. A summary is given in Section 6. This work is useful to estimate the spatiotemporal coupling induced by the dispersive optics in X-ray beamline systems.

## 2. Pulse Propagation by Using Kostenbauder Matrices

In this section, the approach of pulse propagation by using **K** matrices is reviewed [19–21]. We first introduce the definition of **K** matrices in Section 2.1. Then, pulse propagation in real space is discussed by using the generalized Huygens integral in Section 2.2. Finally, pulse propagation in Wigner phase space is introduced in Section 2.3.

### 2.1. Kostenbauder Matrices

To characterize an ultra-short pulse, 6-dimensional vector $\mathbf{V} = (x, \theta_x, y, \theta_y, t, v)^T$ are required. Here, $x$ and $y$ are the transverse coordinates. $\theta_x$ and $\theta_y$ denote the divergences. $t$ and $v$ refer to time and frequency. The optics and free space can be described by $6 \times 6$ **K** matrix.

$$\mathbf{K} = \begin{bmatrix} \frac{\partial x_{\text{out}}}{\partial x_{\text{in}}} & \frac{\partial x_{\text{out}}}{\partial \theta_{\text{xin}}} & 0 & 0 & 0 & \frac{\partial x_{\text{out}}}{\partial v_{\text{in}}} \\ \frac{\partial \theta_{\text{xout}}}{\partial x_{\text{in}}} & \frac{\partial \theta_{\text{xout}}}{\partial \theta_{\text{xin}}} & 0 & 0 & 0 & \frac{\partial \theta_{\text{xout}}}{\partial v_{\text{in}}} \\ 0 & 0 & \frac{\partial y_{\text{out}}}{\partial y_{\text{in}}} & \frac{\partial y_{\text{out}}}{\partial \theta_{\text{yin}}} & 0 & \frac{\partial y_{\text{out}}}{\partial v_{\text{in}}} \\ 0 & 0 & \frac{\partial \theta_{\text{yout}}}{\partial y_{\text{in}}} & \frac{\partial \theta_{\text{yout}}}{\partial \theta_{\text{yin}}} & 0 & \frac{\partial \theta_{\text{yout}}}{\partial v_{\text{in}}} \\ \frac{\partial t_{\text{out}}}{\partial x_{\text{in}}} & \frac{\partial t_{\text{out}}}{\partial \theta_{\text{xin}}} & \frac{\partial t_{\text{out}}}{\partial y_{in}} & \frac{\partial t_{\text{out}}}{\partial \theta_{\text{yin}}} & 1 & \frac{\partial t_{\text{out}}}{\partial v_{\text{in}}} \\ 0 & 0 & 0 & 0 & 0 & 1 \end{bmatrix} = \begin{bmatrix} A_x & B_x & 0 & 0 & 0 & Ex \\ C_x & D_x & 0 & 0 & 0 & Fx \\ 0 & 0 & A_y & B_y & 0 & Ey \\ 0 & 0 & C_y & D_y & 0 & Fy \\ Gx & Hx & Gy & Hy & 1 & I \\ 0 & 0 & 0 & 0 & 0 & 1 \end{bmatrix}. \quad (1)$$

The physical interpretations of the elements in the **K** matrix are summarized in Table 1.

**Table 1.** The physical meaning of each element in the **K** matrix.

| $A_x, A_y$ | $B_x, B_y$ | $C_x, C_y$ |
|---|---|---|
| Transverse magnification | Configuration of the system | Focusing or Defocusing |
| $D_x, D_y$ | $E_x, E_y$ | $F_x, F_y$ |
| Angular magnification | Spatial chirp | Angular dispersion |
| $G_x, G_y$ | $H_x, H_y$ | I |
| Pulse front tilt | Time vs. Angle | Group delay dispersion |

For a system with $n$ elements including free space, the **K** matrix of the system is expressed as

$$\mathbf{K} = \mathbf{K_n} \cdots \mathbf{K_3} \mathbf{K_2} \mathbf{K_1}. \tag{2}$$

If a pulse goes through this system, the output coordinates can be written as a linear combination of the initial coordinates

$$\mathbf{V}_{\text{out}} = \mathbf{K} \mathbf{V}_{\text{in}}. \tag{3}$$

The **K** matrix of free space with length $L$ is given by

$$\mathbf{K}_{\text{free}} = \begin{bmatrix} 1 & L & 0 & 0 & 0 & 0 \\ 0 & 1 & 0 & 0 & 0 & 0 \\ 0 & 0 & 1 & L & 0 & 0 \\ 0 & 0 & 0 & 1 & 0 & 0 \\ 0 & 0 & 0 & 0 & 1 & 0 \\ 0 & 0 & 0 & 0 & 0 & 1 \end{bmatrix}. \tag{4}$$

*2.2. Real Space Propagation*

It is well known that ray tracing in an optical system can be characterized by ordinary **ABCD** matrix, and this is equivalently described by Huygens integral [22]. Similarly, pulse propagation by using **K** matrix can also be presented by the spatiotemporal Huygens integral [19–21]. After going through an optical system which can be described by **K** matrix, the amplitude of the pulse can be expressed as

$$E(x, y, t) = E_0 \exp\left[-i\frac{\pi}{\lambda_0}\begin{pmatrix} x \\ y \\ -t \end{pmatrix}^T \mathbf{Q}^{-1}\begin{pmatrix} x \\ y \\ t \end{pmatrix}\right], \quad \mathbf{Q} = \begin{bmatrix} Q_{11} & Q_{12} & Q_{13} \\ Q_{21} & Q_{22} & Q_{23} \\ Q_{31} & Q_{32} & Q_{33} \end{bmatrix}, \tag{5}$$

where $\lambda_0$ is the central wavelength of the ultra-short pulse. $x, y$ are the lateral position deviation from the pulse center, and $t$ is the time deviation from the pulse time center. The $\mathbf{Q}_{\text{out}}$ matrix at the output can be obtained by transforming $\mathbf{Q}_{\text{in}}$ with the system matrix **K**.

$$
\mathbf{Q}_{\text{out}} = \cfrac{\begin{bmatrix} A_x & 0 & 0 \\ 0 & A_y & 0 \\ G_x & G_y & 1 \end{bmatrix} \mathbf{Q}_{\text{in}} + \begin{bmatrix} B_x & 0 & E_x/\lambda_0 \\ 0 & B_y & E_y/\lambda_0 \\ H_x & H_y & I/\lambda_0 \end{bmatrix}}{\begin{bmatrix} C_x & 0 & 0 \\ 0 & C_y & 0 \\ 0 & 0 & 0 \end{bmatrix} \mathbf{Q}_{\text{in}} + \begin{bmatrix} D_x & 0 & F_x/\lambda_0 \\ 0 & D_y & F_y/\lambda_0 \\ 0 & 0 & 1 \end{bmatrix}}.
\tag{6}
$$

Then, the output pulse is obtained by substituting $\mathbf{Q}_{\text{out}}$ into Equation (5). Actually, it is easy to transform the $\mathbf{Q}$ matrix in $(x, y, v)$, $(\theta_x, \theta_y, v)$ and $(\theta_x, \theta_y, t)$ domains. More details can be found in reference [20]. Pulse propagation in real space by using $\mathbf{K}$ matrices can describe spatiotemporal coupling which is produced by dispersive optical elements, such as blazed grating, asymmetry-cut multilayer, and asymmetry-cut crystal.

### 2.3. Wigner Phase Space Propagation

It is apparent that Equation (5) can not transform to $(x, \theta_x)$, $(y, \theta_y)$ and $(t, v)$ domains, which are called Wigner phase space. Here, we define the distribution of a pulse in Wigner phase space.

$$
W(t, v) = W_0 \exp\left[\begin{pmatrix} t \\ v \end{pmatrix}^T \mathbf{\Omega} \begin{pmatrix} t \\ v \end{pmatrix}\right], \quad \mathbf{\Omega} = \begin{bmatrix} \Omega_{11} & \Omega_{12} \\ \Omega_{21} & \Omega_{22} \end{bmatrix}.
\tag{7}
$$

To propagate in Wigner space, the first step is to obtain the 6-dimensional $\mathbf{K}$ matrix of the system. Then, the $\mathbf{K}$ matrix reduces to a 2-dimensional matrix.

$$
\mathbf{K}_{tv} = \begin{bmatrix} 1 & I \\ 0 & 1 \end{bmatrix}.
\tag{8}
$$

The output $\mathbf{\Omega}_{\text{out}}$ matrix is expressed as

$$
\mathbf{\Omega}_{\text{out}} = \mathbf{K}_{tv}^{-1}{}^T \mathbf{\Omega}_{\text{in}} \mathbf{K}_{tv}^{-1}.
\tag{9}
$$

A chirped pulse in $(t, v)$ phase space can be expressed by using the $\mathbf{\Omega}$ matrix. Similarly, the $\mathbf{\Omega}$ matrices in $(x, \theta_x)$ and $(y, \theta_y)$ domains can be obtained by calculating the $\mathbf{K}_{x\theta_x}$ and $\mathbf{K}_{y\theta_y}$ of the system. We have understood the method of pulse propagation by using $\mathbf{K}$ matrices. In the next section, we investigate the $\mathbf{K}$ matrices of X-ray optics.

## 3. Kostenbauder Matrices of X-ray Optics in XFEL Beamline

The optics generally adopted in soft XFEL beamline systems include mirrors and gratings. Mirrors include plane mirror, cylindrical mirror, spherical mirror, and toroidal mirror, while gratings include plane (VLS) grating, cylindrical (VLS) grating, spherical (VLS) grating and toroidal (VLS) grating. These optics can be described by a unified model: Toroidal VLS grating. In this section, we first derive the unified $\mathbf{K}$ matrix, which can reduce to the $\mathbf{K}$ matrices of the above X-ray optics. Then, we discuss the X-ray optics in different orientations.

### 3.1. Unified Model of X-ray Optics

As shown in Figure 1, an XFEL pulse incidents on a toroidal VLS grating with angle $\alpha$, and reflects with angle $\beta$. The radiuses in the meridian and sagittal directions are $R_m$ and $R_s$, respectively. The groove density $N = N_0(1 + b_2 w)$, where $n_0$, $b_2$, and $w$ are the

central groove density, VLS parameter, and grating coordinate along the meridian direction, respectively. The **ABCD** matrices of plane grating and spherical VLS grating can be found in references [23,24], which can propagate the transverse information of a pulse. Here, we derive the 6-dimensional **K** matrix of Toroidal VLS grating (upward orientation).

$$
\mathbf{K} = \begin{bmatrix}
1 & 0 & 0 & 0 & 0 & 0 \\
-\dfrac{\cos\beta + \cos\alpha}{R_s} & 1 & 0 & 0 & 0 & 0 \\
0 & 0 & -C_{ff} & 0 & 0 & 0 \\
0 & 0 & \dfrac{n_0 b_2 m \lambda_0}{C_{ff} \cos^2\alpha} + \dfrac{1 + C_{ff}}{R_m \cos\alpha C_{ff}} & -\dfrac{1}{C_{ff}} & 0 & -\dfrac{n_0 m \lambda_0^2}{\cos\beta c} \\
0 & 0 & \dfrac{n_0 m \lambda_0}{c \cos\alpha} & 0 & 1 & 0 \\
0 & 0 & 0 & 0 & 0 & 1
\end{bmatrix},
\tag{10}
$$

where $c$, $\lambda_0$, and $m$ are the speed of light, central wavelength, and diffraction order, respectively. $C_{ff} = \cos\beta / \cos\alpha$ and is called the asymmetry parameter. In Appendix A, we provide more detailed derivations of Equation (10).

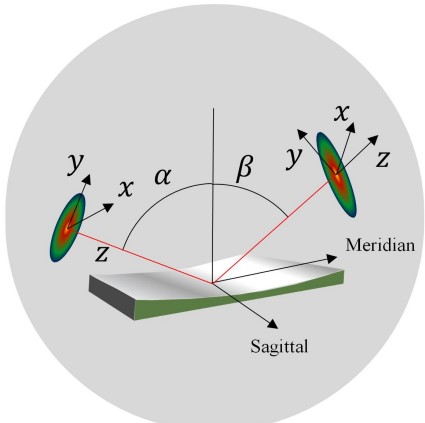

**Figure 1.** Schematic illustration of the unified optics model: Toroidal VLS grating.

Equation (10) can almost reduce to the **K** matrices of all the optical elements in soft XFEL beamline by adjusting the parameters in the **K** matrix of Toroidal VLS grating. In Table 2, the corresponding parameters of different kinds of optics are summarized.

**Table 2.** Parameters of different types of optics. Here, ✓ denotes the relevant parameter remaining unchanged. For gratings with constant spacing, we only need to set $b_2 = 0$.

| | $R_m$ | $R_s$ | $C_{ff}$ | $b_2$ | $n_0$ |
|---|---|---|---|---|---|
| Toroidal VLS grating | ✓ | ✓ | ✓ | ✓ | ✓ |
| Spherical VLS grating | $R_m = R_s$ | | ✓ | ✓ | ✓ |
| Cylindrical VLS grating | ✓ | ∞ | ✓ | ✓ | ✓ |
| Plane VLS grating | ∞ | ∞ | ✓ | ✓ | ✓ |
| Toroidal grating | ✓ | ✓ | ✓ | 0 | ✓ |
| Spherical grating | $R_m = R_s$ | | ✓ | 0 | ✓ |
| Cylindrical grating | ✓ | ∞ | ✓ | 0 | ✓ |

**Table 2.** *Cont.*

|  | $R_m$ | $R_s$ | $C_{ff}$ | $b_2$ | $n_0$ |
|---|---|---|---|---|---|
| Plane grating | ∞ | ∞ | ✓ | 0 | ✓ |
| Toroidal mirror | ✓ | ✓ | 1 | 0 | 0 |
| Spherical mirror | $R_m = R_s$ | | 1 | 0 | 0 |
| Cylindrical mirror | ✓ | ∞ | 1 | 0 | 0 |
| Plane mirror | ∞ | ∞ | 1 | 0 | 0 |

### 3.2. Kostenbauder Matrices in Different Orientations

Generally, the X-ray optics in XFEL beamline are oriented in different directions, including upward, downward, leftward, and rightward directions. Equation (10) is the **K** matrix in upward orientation. The matrices in other orientations can be obtained by

$$\mathbf{K} = \mathbf{R}^{-1}\mathbf{K}_{up}\mathbf{R}, \tag{11}$$

where **R** is the transformation matrix of coordinate. The transformation matrices are given in Equation (12). By substituting the transformation matrices $\mathbf{R}_{down}$, $\mathbf{R}_{left}$, and $\mathbf{R}_{right}$ into Equation (11), the matrix $\mathbf{K}_{up}$ can be transformed to downward, leftward, and rightward orientations, respectively.

$$
\underbrace{\begin{bmatrix}
-1 & 0 & 0 & 0 & 0 & 0 \\
0 & -1 & 0 & 0 & 0 & 0 \\
0 & 0 & -1 & 0 & 0 & 0 \\
0 & 0 & 0 & -1 & 0 & 0 \\
0 & 0 & 0 & 0 & 1 & 0 \\
0 & 0 & 0 & 0 & 0 & 1
\end{bmatrix}}_{\mathbf{R}_{down}},
\underbrace{\begin{bmatrix}
0 & 0 & -1 & 0 & 0 & 0 \\
0 & 0 & 0 & -1 & 0 & 0 \\
1 & 0 & 0 & 0 & 0 & 0 \\
0 & 1 & 0 & 0 & 0 & 0 \\
0 & 0 & 0 & 0 & 1 & 0 \\
0 & 0 & 0 & 0 & 0 & 1
\end{bmatrix}}_{\mathbf{R}_{left}},
\underbrace{\begin{bmatrix}
0 & 0 & 1 & 0 & 0 & 0 \\
0 & 0 & 0 & 1 & 0 & 0 \\
-1 & 0 & 0 & 0 & 0 & 0 \\
0 & -1 & 0 & 0 & 0 & 0 \\
0 & 0 & 0 & 0 & 1 & 0 \\
0 & 0 & 0 & 0 & 0 & 1
\end{bmatrix}}_{\mathbf{R}_{right}}. \tag{12}
$$

## 4. Examples of Application

Having obtained the **K** matrices of X-ray optics, we can calculate pulse propagation in X-ray optical systems. In this section, we apply **K** matrices to estimate pulse properties through the grating monochromator and grating compressor.

### 4.1. Grating Monochromator in FEL Beamline

In this subsection, we first briefly introduce a typical VLS-PGM that may be adopted in soft XFEL beamlines. Then, we apply **K** matrices to investigate the spatiotemporal coupling of an ultra-short Gaussian pulse through the monochromator. The configuration of VLS-PGM is shown in Figure 2. It consists of a plane pre-mirror, a VLS plane grating, and an exit slit. The function of the pre-mirror is to change the incident angle of the grating and keep the offset of the beamline constant. The VLS plane grating can focus the pulse at the exit slit and disperse different wavelength components. The exit slit is used to filter the spatial dispersive pulse, then a monochromatic pulse is obtained.

For the design of VLS-PGM in the XFEL beamline, we hope to obtain high resolving power and tiny pulse stretching. Therefore, the estimations of pulse widths after the grating and at the exit slit are required. Here, we apply the optical matrix approach to simulate the monochromator. The distance $l$ from the source to the pre-mirror is 206.5 m. The distance $d$ between the pre-mirror and grating is 0.5 m. The distance from the grating to the exit slit is 118 m. The grazing angle of the pre-mirror is 1.313°. The parameters of source and grating are summarized in Table 3. The **K** matrix of the VLS-PGM system is given by

$$\mathbf{K}_{mono} = \mathbf{K}_{free}^{r'}\mathbf{K}_{down}^{grating}\mathbf{K}_{free}^{d}\mathbf{K}_{up}^{mirror}\mathbf{K}_{free}^{l}, \tag{13}$$

where $\mathbf{K}_{\text{free}}^{r'}$, $\mathbf{K}_{\text{down}}^{\text{grating}}$, $\mathbf{K}_{\text{free}}^{d}$, $\mathbf{K}_{\text{up}}^{\text{mirror}}$ and $\mathbf{K}_{\text{free}}^{l}$ are the **K** matrices of free space with distance $r'$, VLS plane grating in downward orientation, free space between pre-mirror and grating, pre-mirror in upward orientation and free space with distance $l$, respectively. These matrices can be obtained by using the method in Equations (10)–(12). The spatiotemporal response of VLS-PGM can be calculated by using the model of pulse propagation in real space.

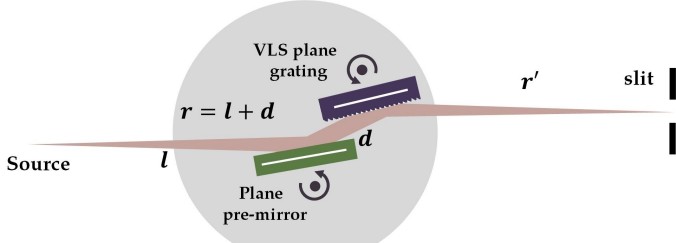

**Figure 2.** Schematic illustration of the VLS-PGM. Here, $r$ is the distance from the source point to the grating, and $r'$ is the distance from the grating to the exit slit.

**Table 3.** Simulation parameters of VLS-PGM.

| Source Parameters | | | |
|---|---|---|---|
| $\lambda_0$ [nm] | $\sigma_x$ [μm] | $\sigma_y$ [μm] | $\sigma_t$ [fs] |
| 1 | 30.25 | 30.25 | 29.73 |
| Grating parameters | | | |
| $n_0$ [1/m] | $b_2$ [1/m] | $\alpha$ [°] | $\beta$ [°] |
| $3 \times 10^5$ | $2.882 \times 10^{-2}$ | 89.062 | 88.312 |

Before the VLS plane grating, the pulse intensity distribution in the $(y, t)$ domain is shown in Figure 3a. The pulse intensity distribution after the grating is shown in Figure 3b, and we can observe the pulse front tilt effect. Mathematically, the pulse widths after the VLS plane grating can be described by

$$\sigma_{yb} = \sigma_{ya}C_{ff}, \tag{14}$$

$$\sigma_{tb}^2 = \sigma_{ta}^2 + (G_y\sigma_{ya})^2, \tag{15}$$

where $\sigma_{ya}$ and $\sigma_{ta}$ are the r.m.s of intensity in the vertical dimension and pulse duration before the VLS plane grating, respectively. $\sigma_{yb}$, $\sigma_{tb}$ are r.m.s of intensity in the vertical dimension and pulse duration after the VLS plane grating, respectively. Equation (15) shows that the pulse duration after the VLS plane grating is the convolution of the initial pulse duration and the stretched term induced by the grating.

The pulse intensity distribution after propagating 59 m from the grating is presented in Figure 3d. It is obvious that pulse rotation occurs due to the focusing effect of VLS grating. In the focus, as shown in Figure 3e, we can find that the pulse is focused in transverse, and the pulse duration is stretched.

Here, we use Shadow and SRW to benchmark the approach of pulse propagation by using **K** matrices. We also compared pulse propagation using the TDMD method and the **K**-matrix method. Figure 3c shows the calculation of the horizontal intensity distribution at the focus (exit slit). The simulation results indicate that the **K**-matrix approach makes a great agreement with Shadow SRW, and TDMD. Figure 3f shows the vertical intensity distribution at the focus. We can find that the intensity profile calculated by the **K**-matrix approach is broader than the results calculated by Shadow and SRW, and the results calculated by **K**-matrix method exhibit a high degree of concordance with those estimated by the TDMD method. This is because simulations performed by Shadow and SRW are under the assumption of a single wavelength and can not describe the

spatiotemporal couplings. However, the methods using **K** matrices and TDMD can describe pulse propagation which includes temporal distribution and spectral information(Fourier transform limited bandwidth). That means different wavelength components are dispersed in the focus of the VLS grating, and the intensity width is undoubtedly larger than the results calculated by Shadow and SRW. Mathematically, the pulse widths at the focus can be expressed as

$$\sigma_{yd}^2 = (\sigma_{ys}\frac{r'}{rC_{ff}})^2 + (\frac{F_y r'}{4\pi\sigma_{ts}})^2, \tag{16}$$

$$\sigma_{td} = \sigma_{tb}, \tag{17}$$

where $\sigma_{ys}$ and $\sigma_{ts}$ are the r.m.s. of intensity width in the vertical dimension and pulse duration at the source point, respectively. $\sigma_{yd}$ and $\sigma_{td}$ are the r.m.s. of intensity width in the vertical dimension and pulse duration at the focus, respectively. The first term in Equation (16) is corresponding to the magnification (demagnification) and agrees with the results estimated by Shadow and SRW. The second term in Equation (16) is corresponding to dispersion. In Figure 3f, the r.m.s. of the intensity calculated by **K** matrix is 14.37 µm, and that of the intensity calculated by Shadow and SRW is around 9.58 µm, which is closed to demagnification term $\sigma_{ys}r'/rC_{ff} = 9.57$ µm. The dispersion term $F_y r'/4\pi\sigma_{ts} = 10.71$ µm. Certainly, we have $\sigma_{yd} = \sqrt{9.57^2 + 10.71^2}$ µm = 14.37 µm. Compared with the TDMD method, the **K**-matrix method offers a more simple computational process and demands comparatively fewer computational resources.

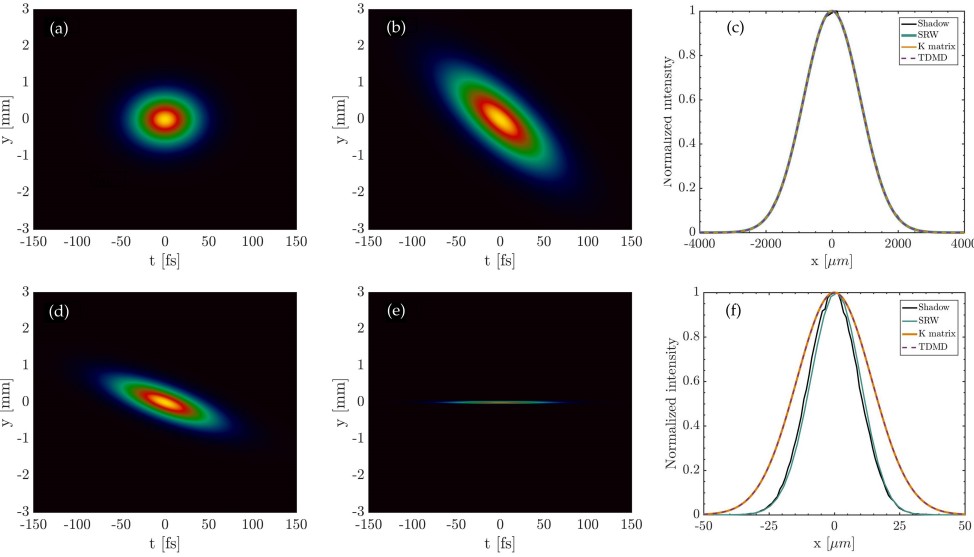

**Figure 3.** The pulse distributions in $(y, t)$ domain at different positions. (**a**) Before the VLS grating, $\sigma_y$ and $\sigma_t$ are 544.39 µm and 29.73 fs. (**b**) After the VLS grating, $\sigma_y$ and $\sigma_t$ are 981.09 µm and 44.64 fs. (**d**) After propagating 59 m from the VLS grating, $\sigma_y$ and $\sigma_t$ are 490.71 µm and 44.64 fs. (**e**) At the focus (exit slit), $\sigma_y$ and $\sigma_t$ are 14.37 µm and 44.64 fs. (**c**) At the focus, the transverse intensity profiles estimated by Shadow, SRW, TDMD, and **K** matrix. (**f**) At the focus, the vertical intensity profiles calculated by Shadow, SRW, TDMD, and **K** matrix. Here, the transverse (vertical) intensity is the projected intensity of the 2-D contour plot in the transverse (vertical) coordinate.

### 4.2. Pulse Compressor in XFEL

In this subsection, the application of optical matrix in a soft XFEL pulse compressor is studied. We first establish the **K** matrix of a typical negative group delay dispersion (GDD) grating compressor. Then we calculate an up-chirped XFEL pulse going through the grating compressor.

Figure 4 is a typical negative GDD grating compressor that can be used to compress up-chirped soft X-ray FEL pulses. The compressor consists of two identical blazed grating

$G_1$, $G_2$, and the distance between $G_1$ and $G_2$ is $N$. The incident angles of $G_1$ and $G_2$ are $\alpha_1$ and $\alpha_2$, while the diffraction angles are $\beta_1$ and $\beta_2$. The diffraction orders of $G_1$ and $G_2$ are 1 and $-1$, respectively. By adjusting the distance $N$, we can change the GDD of the compressor Table 4.

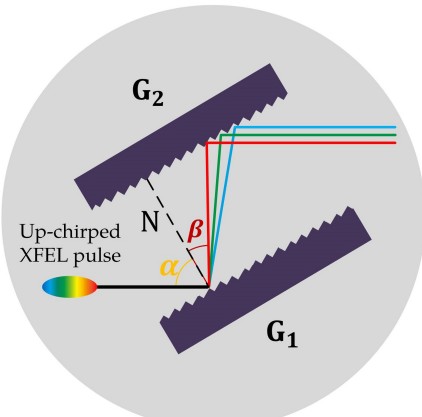

**Figure 4.** Schematic illustration of negative GDD grating compressor.

**Table 4.** Parameters of negative GDD grating compressor.

| Distance | $G_1$ $G_2$ | G1 (m = +1) | | G2 (m = −1) | |
|---|---|---|---|---|---|
| $N$ [m] | $n_0$ [1/m] | $\alpha_1$ [°] | $\beta_1$ [°] | $\alpha_2$ [°] | $\beta_2$ [°] |
| $2.167 \times 10^{-2}$ | $1.2 \times 10^6$ | 88.37 | 84.87 | 84.87 | 88.37 |

The **K** matrix of the compressor is given by

$$\mathbf{K}_{\text{compressor}} = \mathbf{K}_{\text{down}}^{G_2}\mathbf{K}_{\text{free}}\mathbf{K}_{\text{up}}^{G_1}, \tag{18}$$

where $\mathbf{K}_{\text{dow}}^{G_2}$, $\mathbf{K}_{\text{free}}$, and $\mathbf{K}_{\text{up}}^{G_1}$ are the **K** matrices of $G_2$ in downward orientation, free space with distance $N/\cos\beta_1$ and $G_1$ in upward orientation, respectively. The pulse compression can be calculated by using $\mathbf{\Omega}$ matrix pulse propagation in Wigner phase space.

In this calculation, the FWHM of the bandwidth of the up-chirped soft X-ray FEL pulse is 0.88%. The central wavelength and the FWHM of pulse duration are 3 nm and 10 fs, respectively. The simulation parameters of the negative GDD compressor are given in Table 4. The Wigner distribution of the input pulse is shown in Figure 5a, and the Wigner distribution of the output pulse is shown in Figure 5b. For **K**-matrix method, the 1-D spectral and temporal distribution in Figure 5c,d is obtained by projecting the 2-D Wigner distribution in Figure 5b onto the photon energy axis and time axis, respectively. We can observe that the pulse duration changes from 10 fs to 0.5 fs. As shown in Figure 5c,d, we compared the **K**-matrix method and the TDMD method, and the calculation shows that the simulation results of these two methods are highly consistent.

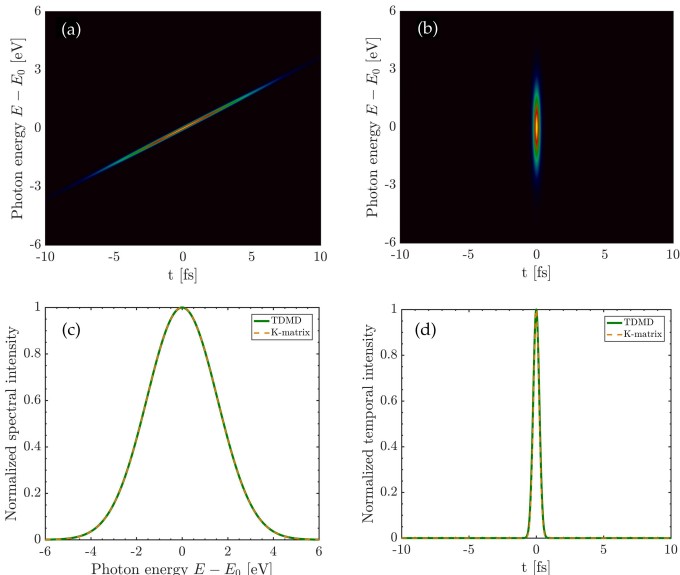

**Figure 5.** (**a**) The Wigner distribution of the up-chirped pulse before the compressor. (**b**) The Wigner distribution of the pulse after the compressor. (**c**) The normalized spectral intensity distribution after the compressor. (**d**) The normalized temporal intensity distribution after the compressor.

## 5. Discussion

The simulation of XFEL pulse propagation is a key process in beamline design. Several simulation packages are usually adopted, such as Shadow [11], SRW [12], HYBRID [13], xrt [14], and MOI [15]. The above software can be used to simulate steady-state beam propagation, where the beam does not have temporal distribution. These software tools can provide a good evaluation of the transverse beam propagation characteristics in non-dispersive optical systems, but cannot simulate the spatiotemporal coupling effects in dispersive optical systems. The TDMD method [16–18] and the **K**-matrix method [19–21] can be classified as time-dependent beam propagation, where the beam is pulsed and has temporal distribution. The two methods can not only describe the transverse propagation characteristics of X-ray pulses very well but also accurately simulate the spatiotemporal coupling effects generated in dispersive optical systems, such as spatial chirp, angular dispersion, pulse front tilt, and so on. Compared to the TDMD method, the K-matrix method has the advantages of simpler calculation and less dependence on sampling.

This work extended the method of pulse propagation by using **K** matrices to X-ray regime and derived the **K** matrices of different types of X-ray optics. We applied our method to simulate two typical dispersive X-ray systems: a grating monochromator and a grating compressor. By using this approach, the spatiotemporal properties of ultra-short X-ray pulses in dispersive systems can be well simulated. Our approach and the existing simulation tools [11–15] demonstrate good complementarity in beamline design and optimization. Based on the characteristics of the various simulation tools mentioned above, we can choose simulation tools according to our needs in beamline design and simulation.

## 6. Conclusions

In this paper, we derived a unified 6-dimensional **K** matrix to describe the optics in soft XFEL beamline. This unified matrix can reduce to different types of X-ray optics, including plane mirror, cylindrical mirror, spherical mirror, toroidal mirror, plane grating, cylindrical grating, spherical grating and toroidal grating, plane VLS grating, cylindrical VLS grating, spherical VLS grating, and toroidal VLS grating. By using this method, we can simulate X-ray pulse propagation in real space and Wigner phase space. We

successfully applied this method to calculate pulse propagation through a VLS-PGM. The simulation benchmark was performed by using Shadow and SRW, and the results make great agreement with our method. We also found some differences which arise from different assumptions. Simulations of Shadow and SRW are carried out under the assumption of a single wavelength, while our method of pulse propagation includes spectral information which corresponds to a Fourier transform limited bandwidth. We also discussed the application in the simulation of a grating compressor.

**Author Contributions:** Conceptualization, Q.W., W.Z. and C.Y.; Funding acquisition, C.Y.; Investigation, K.H. and C.Y.; Methodology, Y.Z. and Z.X.; Software, K.H. and Y.Z.; Supervision, Q.W., W.Z. and C.Y.; Validation, Y.Z.; Writing–original draft, K.H.; Writing—review and editing, Q.W., W.Z. and C.Y. All authors have read and agreed to the published version of the manuscript.

**Funding:** This research was funded by the National Natural Science Foundation of China (Grant No. 12005135 and 22288201) and National Key R&D Program of China (Grant No. 2018YFE0203000) and Scientific Instrument Developing Project of the Chinese Academy of Sciences (Grant No. GJJSTD20190002).

**Institutional Review Board Statement:** Not applicable.

**Informed Consent Statement:** Not applicable.

**Data Availability Statement:** Not applicable.

**Conflicts of Interest:** The authors declare no conflict of interest.

## Appendix A. The Derivation of the Unified Model

An X-ray beam can be regarded as composed of a group of particles with different coordinates, and the state of these particles can be described by $(x, y, t, \theta_x, \theta_y, v)$. The reference particle's coordinates are located at the origin, and its propagation path is called the optical axis. Specifically, $x, y, t$ are the spatial coordinates, describe the position of the particle relative to the reference particle. The positive direction of the coordinate axis $t$ always corresponds to the direction of reference particle propagation. The coordinate axes $x$ and $y$ are perpendicular to $t$ and are used to describe the lateral position. As shown in Figure A1a , The spatial coordinate system of the X-ray beam and the coordinate system of the toroidal VLS grating are independent of each other. $\theta_x, \theta_y$ describe the angle of the particle's propagation direction relative to the reference particle, and $v$ represents the frequency difference between the particle and the reference particle. The coordinate system of the toroidal VLS grating is composed of the meridian direction $M$, the sagittal direction $S$, and the normal direction $N$ of the optical element surface. The grooves of toroidal VLS grating is perpendicular to the meridian direction.

The unified optics model is nominally a VLS toroidal grating. To obtained the **K** matrix of this unified model, we need to make the following assumptions:

- Paraxial approximation: $\theta_x$ and $\theta_y$ are small.
- The radius of curvature $R_M$ and $R_S$ and are much larger than the beam size.

Based on the definitions and assumptions mentioned above, we will derive the **K** matrix of the toroidal VLS grating in the following subsections. The derivation process is decomposed into the sagittal dimension, meridional dimension, and time dimension. The optical ray mentioned in the following is the propagation path of the particle.

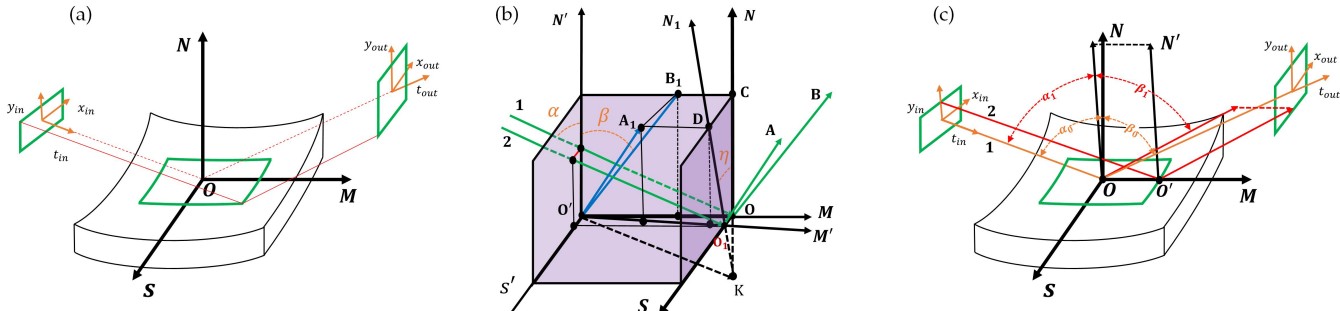

**Figure A1.** (**a**) Illustration of the coordinate system. (**b**) Illustration of ray propagation in the sagittal dimension. (**c**) Illustration of ray propagation in the meridian dimension.

*Appendix A.1. Sagittal Dimension*

In sagittal dimension, it is easy to obtain that the transverse magnification $\partial x_{\text{out}}/\partial x_{in}$ and the angular magnification $\partial \theta_{\text{xout}}/\partial \theta_{xin}$ are 1. Here, we focus on the deviation of the focusing term $\partial \theta_{\text{xout}}/\partial x_{in}$. In Figure A1b, we present two rays (green curves) going through the toroidal VLS grating, where both rays are incident in a parallel manner. Ray 1 is the reference ray, which is incident at point $O$, and both the incident and reflected rays are within the meridian plane $NOM$. Ray 2 is incident at point $O_1$, which is offset from $O$ by a distance $S$ in the sagittal direction. As the toroidal VLS grating has a curvature of $R_S$ in the sagittal direction, The plane spanned by the incident and reflected vectors of ray 2 is inclined to the meridian plane by an angle $\eta$. The normals at points $O$ and $O_1$ intersect at point $K$. $O'$ is a point on the $M$-axis and $O'K$ is the unit vector in the direction of the incident ray. $O'A_1$ and $O'B_1$ are the unit vectors of the reflected rays, $O'A$ and $O'B$, respectively. Points $C$ and $D$ are the orthogonal projections of points $B'$ and $A'$ onto the lines $ON$ and $O_1N_1$, respectively. According to the assumptions postulated previously, the quadrilateral $A'B'CD$ can be approximated as a rectangle, and hence, it follows that the length of segment $\overline{A_1B_1}$ is approximately equivalent to that of $\overline{CD}$. Therefore, we can obtain the angle by which the reflected ray deviates from the direction of the reference ray.

$$\theta_{\text{xout}} \approx \frac{\overline{A_1B_1}}{\overline{O'A_1}} \approx \overline{CD} \approx \eta \overline{CK} = \frac{S}{R_S}(\cos\alpha + \cos\beta), \tag{A1}$$

where, $\eta = S/R_S$, and $\overline{CK} = \overline{O'K}\cos\alpha + \overline{O'B_1}\cos\beta = \cos\alpha + \cos\beta$. Based on the definition of coordinate system, we have

$$\frac{\partial \theta_{\text{xout}}}{\partial x_{\text{in}}} = \pm \frac{\cos\alpha + \cos\beta}{R_s}. \tag{A2}$$

For optical elements possessing a concave surface in the sagittal plane, a positive deviation $S$ in the sagittal direction of the incident point relative to the reference incident point causes a negative angular deviation in the reflected ray. Conversely, for optical elements possessing a convex surface in the sagittal plane, a positive deviation $S$ in the sagittal direction of the incident point relative to the reference incident point results in a positive angular deviation in the reflected ray. In Equation (8), the toroidal VLS grating is concave in both the meridian and sagittal directions, so a negative sign is used.

*Appendix A.2. Meridian Dimension*

In Figure A1c, two parallel rays are incident on the toroidal VLS grating. Ray 1 is the reference ray, which is incident at point $O$, while Ray 2 is incident at point $O'$, which is offset from $O$ by a distance $M$ in the meridian direction. Due to the curvature of the optical element in the meridian direction, which is denoted as $R_m$, and the variation of the grating

line density in the meridian direction, the reflected beam of Ray 2 will have an angular difference with respect to the reference ray. The grating equation is given by

$$\sin \alpha + \sin \beta = n(M)m\lambda, \quad n(M) = n_0(1 + b_2 M). \tag{A3}$$

Then, we have

$$\Delta \beta = \frac{\Delta n m \lambda}{\cos \beta} - \Delta \alpha \frac{\cos \alpha}{\cos \beta}, \tag{A4}$$

where

$$\Delta \alpha = \frac{M}{R_M}, \quad \Delta n = M n_0 b_2. \tag{A5}$$

The angle of the reflected ray deviation from the reference ray can be expressed as

$$\theta_{\text{yout}} = \Delta \beta - \Delta \alpha = \frac{M n_0 b_2 m \lambda}{\cos \beta} - \frac{M(1 + C_{ff})}{R_M C_{ff}}. \tag{A6}$$

where, the deviation $M = -y_{\text{in}} / \cos \alpha$. Thus,

$$\frac{\partial \theta_{\text{yout}}}{\partial y_{\text{in}}} = \frac{n_0 b_2 m \lambda}{\cos \alpha^2 C_{ff}} + \frac{1 + C_{ff}}{R_M \cos \alpha C_{ff}}. \tag{A7}$$

The transverse magnification can be written as

$$\frac{\partial y_{\text{out}}}{\partial y_{\text{in}}} = -\frac{\overline{OO'} \cos \beta}{\overline{OO'} \cos \alpha} = -C_{ff}. \tag{A8}$$

The angular magnification and angular dispersion can be derived from Equation (A3)

$$\frac{\partial \theta_{\text{yout}}}{\partial \theta_{\text{yin}}} = \frac{\Delta \beta}{\Delta \alpha} = -\frac{\cos \alpha}{\cos \beta} = -\frac{1}{C_{ff}}, \quad \frac{\partial \theta_{\text{yout}}}{\partial v_{\text{in}}} = \frac{\Delta \beta}{\Delta v} = -\frac{n_0 m \lambda^2}{\cos \beta c}. \tag{A9}$$

*Appendix A.3. Time Dimension*

As is well known, the pulse duration of an ultrashort pulse will be stretched after passing through a grating, and the stretching term can be expressed as

$$\tau = \frac{\mathcal{N} m \lambda}{c} \approx \frac{W n_0 m \lambda}{c} = y_{\text{in}} \frac{n_0 m \lambda}{c \cos \alpha}, \tag{A10}$$

where $\mathcal{N}$ is the number of grooves covered by the footprint of the incident beam. Thus the term corresponding to pulse front tilt is given by

$$\frac{\partial t_{\text{out}}}{\partial y_{\text{in}}} = \frac{n_0 m \lambda}{c \cos \alpha}. \tag{A11}$$

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
