# Peer review of "Ultrashort X-ray Free Electron Laser Pulse Manipulation by Optical Matrix"

_photonics, doi:10.3390/photonics10050491_

Round 1

Reviewer 1 Report

The manuscript by Kai Hu et al considers the 6 x 6 matrix description of Gaussian pulse propagation through a general case of VLS grating. The novelty of the manuscript is in treating the temporal properties of the pulses -- namely, ultrashort pulses and related effects, such as chirp and pulse compression. This study is timely and actual since several XFEL facilities can provide corresponding pulses and considered VLS gratings are also available. To predict the evolution of such pulses in VLS gratings based on analytical expressions would be helpful.
The manuscript is overall clearly written, the presented examples are illustrative and interesting. However, I have two issues that, in my opinion, should be elaborated on further:

1. The key theoretical result of the manuscript -- eq. (10) -- is given without derivation. The derivation steps should be outlined in the main text, and the details of derivation should be given in an Appendix.

2. The comparison with wavefront-propagation codes presented in Figure 3 is very helpful and I agree with the discussion between Eqs. (14a,b) and (15a,b). However, thanks to the linearity of considered optical elements, the wavefront-propagation codes can be applied to each frequency of the beam, and then obtained amplitudes can be summed up. In this way, for Gaussian beams, the obtained results should coincide with the approach based on Kostenbauder matrices. In my opinion, such a calculation should be added to the comparison.

There are also a few minor comments:
-the sentences in lines 180 and 187 are almost repeated;
-in reference 23, Yuri is the given name, S. stands for Shvyd'ko which is the family name.

Summing up, after the two issues described above (derivation of Eq. (10) and comparison to frequency-summed wave-propagation simulations) will be resolved, I will be happy to see the manuscript published.

Author Response

Dear Reviewer,

Thank you very much for taking the time to review our manuscript. We appreciate your thoughtful comments and suggestions, which have helped to improve the quality of our work. Your expertise and insights were invaluable to us, and we are grateful for the time and effort you have devoted to this review process. Some of your questions were answered below.

Reviewer:

The manuscript by Kai Hu et al considers the 6 x 6 matrix description of Gaussian pulse propagation through a general case of VLS grating. The novelty of the manuscript is in treating the temporal properties of the pulses -- namely, ultrashort pulses and related effects, such as chirp and pulse compression. This study is timely and actual since several XFEL facilities can provide corresponding pulses and considered VLS gratings are also available. To predict the evolution of such pulses in VLS gratings based on analytical expressions would be helpful.

  1. Reviewer’s Comments: The key theoretical result of the manuscript -- eq. (10) -- is given without derivation. The derivation steps should be outlined in the main text, and the details of derivation should be given in an Appendix.

Author’s Response: We have added the derivation process in the revised manuscript.

  1. Reviewer’s Comments: The comparison with wavefront-propagation codes presented in Figure 3 is very helpful and I agree with the discussion between Eqs. (14a,b) and (15a,b). However, thanks to the linearity of considered optical elements, the wavefront-propagation codes can be applied to each frequency of the beam, and then obtained amplitudes can be summed up. In this way, for Gaussian beams, the obtained results should coincide with the approach based on Kostenbauder matrices. In my opinion, such a calculation should be added to the comparison.

Author’s Response: This is a very good suggestion, and our viewpoint is consistent with the reviewer's. In principle, we can use the method of wavefront propagation to propagate pulses of different frequencies through the optical system separately and then recombine them. This method is essentially mode decomposition, which requires knowing the temporal structure of the pulse. Unfortunately, the wavefront propagation software SRW does not have the function of mode decomposition. We have also studied the mode decomposition method for propagating ultrashort pulses in another work, and the simulation results of the mode decomposition method are consistent with the results obtained using the matrix propagation method employed in this paper. We have added a comparison between the two methods in the revised manuscript.

  1. Reviewer’s Comments: There are also a few minor comments:

-the sentences in lines 180 and 187 are almost repeated;

-in reference 23, Yuri is the given name, S. stands for Shvyd'ko which is the family name.

Author’s Response: We have deleted the repeated sentence and modified the errors in reference.

Reviewer 2 Report

The manuscript describes the formalism of transfer matrices, representative of optical systems in x-rays, and their applications to two significant examples.

The manuscript is scientifically sound, some minor revision is required before its publication. Please find my criticisms and questions below.

1. Figure 3, it is not clear what "transverse intensity" and "vertical intensity" refer to. Please clarify the axis on which the 2-D contour plot is projected in order to produce plots (c) and (f).

2. Since the discrepancy in fig.3(f) between the methods is attributed to the monochromatic assumption of SRW and Shadow, the authors should clarify if such codes are capable of ray-tracing including a finite bandwidth. If so, the comparison should be done with this option turned on. If not, I encourage the authors to highlight, in the Discussion section, pros and cons of the proposed matrix method compared to the existing and more sophisticated codes (e.g., diffraction, evaluation of coherence, transmission efficiency, etc.). When do the authors recommend to use those codes? When the matrix formalism?

3. I would have expected a similar comparison of methods for the case in fig.4. Is there not any other code which is suitable to the modelling of dispersive lattices, to be compared with the matrix method? Please comment and expand.

Author Response

Dear reviewer:

We gratefully thank you for the precious time the reviewer spent making constructive remarks. Some of your questions were answered below.

Reviewer:

The manuscript describes the formalism of transfer matrices, representative of optical systems in x-rays, and their applications to two significant examples. The manuscript is scientifically sound, some minor revision is required before its publication. Please find my criticisms and questions below.

  1. Reviewer’s Comments: Figure 3, it is not clear what "transverse intensity" and "vertical intensity" refer to. Please clarify the axis on which the 2-D contour plot is projected in order to produce plots (c) and (f).

   Author’s Response: The transverse (vertical) intensity is the projected intensity of the 2-D contour plot in the transverse (vertical) coordinate. We have clarified this in the revised manuscript.

  1. Reviewer’s Comments: Since the discrepancy in fig.3(f) between the methods is attributed to the monochromatic assumption of SRW and Shadow, the authors should clarify if such codes are capable of ray-tracing including a finite bandwidth. If so, the comparison should be done with this option turned on. If not, I encourage the authors to highlight, in the Discussion section, pros and cons of the proposed matrix method compared to the existing and more sophisticated codes (e.g., diffraction, evaluation of coherence, transmission efficiency, etc.). When do the authors recommend to use those codes? When the matrix formalism?

Author’s Response: The source of SHADOW can be set to a finite bandwidth, but SHADOW can not calculate the spatiotemporal coupling due to a lack of temporal distribution. SRW can not calculate wave propagation with a finite bandwidth. The advantage of our method is that this approach is convenient to estimate dispersive systems. We have added a discussion in the revised manuscript.

  1. Reviewer’s Comments: I would have expected a similar comparison of methods for the case in fig.4. Is there not any other code which is suitable to the modelling of dispersive lattices, to be compared with the matrix method? Please comment and expand.

Author’s Response:  So far, SHADOW and SRW can not estimate pulse propagation in Wigner phase space. In principle, the method of mode decomposition is capable to calculate spatiotemporal coupling in real space. We have added the comparison between the method of mode decomposition and the projected intensity of the 2-D Wigner distribution in the revised manuscript.

Reviewer 3 Report

In this study, the authors utilized K matrices to model the behavior of ultra-short x-ray pulses in the x-ray regime and derived the K matrices for various types of x-ray optics. They applied this approach to investigate the spatiotemporal properties of ultra-short x-ray pulses in two dispersive x-ray systems: a grating monochromator and a grating compressor. The simulations produced highly accurate results that were consistent with those obtained from Shadow and SRW, making this research highly relevant and scientifically significant. The paper is well-structured and easy to follow. Although experimental data is not yet available to validate the simulations, the authors could potentially strengthen their findings by providing such data.

Author Response

Dear reviewer:

We would like to thank you for your careful reading, helpful comments, and constructive suggestions, which has significantly improved the presentation of our manuscript.

Reviewer:

Reviewer’s Comments: In this study, the authors utilized K matrices to model the behavior of ultra-short x-ray pulses in the x-ray regime and derived the K matrices for various types of x-ray optics. They applied this approach to investigate the spatiotemporal properties of ultra-short x-ray pulses in two dispersive x-ray systems: a grating monochromator and a grating compressor. The simulations produced highly accurate results that were consistent with those obtained from Shadow and SRW, making this research highly relevant and scientifically significant. The paper is well-structured and easy to follow. Although experimental data is not yet available to validate the simulations, the authors could potentially strengthen their findings by providing such data.

Author’s Response: Thanks for your comments. Currently, we do not have experimental data yet, but in our paper, we have made sufficient comparisons with other simulation software and algorithms, all of which have been confirmed in experiments.

Round 2

Reviewer 1 Report

The corrections to the manuscript "Ultrashort x-ray free electron laser pulse manipulation by optical matrix" have essentially improved the manuscript and addressed all the issues that were raised in the previous review. The comparison with the TDMD method is very convincing and adds valuable confirmation to the presented method. The added Appendix A is very helpful, I find it to be an important part of the paper. I do not have any comments regarding parts A.2, A.3. Unfortunately, I could not completely follow and completely understand the derivation presented in part A.1. In this regard, I have a comment:

1. The points A_1, B_1, C, D shown in Figure A1 are not explained in the text. Similarly, plane A_1B_1CD is not defined. Because of this, the equation (A1) is not self-evident. In particular, it was not clear to me why CK = cos \alpha + cos \beta. A reason for this is that point C was not defined in the text.

Summing up, there is one issue with the explanation in Appendix A.1 that should be resolved before the final publication. Most likely, just describing the points A_1, B_1, C, D shown in Figure A1 could be sufficient to make equation (A1) self-evident. Otherwise, the manuscript, in my opinion, is ready to be published.

Author Response

Dear Reviewer,

I wanted to express my gratitude for your valuable feedback on my manuscript. Your comments and suggestions helped me to improve the quality of my work, and I appreciate the time and effort you took to review it. Our response is presented in the attached file.

Sincerely

Chuan Yang
